# Association of Urinary and Plasma Levels of Trimethylamine N-Oxide (TMAO) with Foods

**DOI:** 10.3390/nu13051426

**Published:** 2021-04-23

**Authors:** Mauro Lombardo, Giovanni Aulisa, Daniele Marcon, Gianluca Rizzo, Maria Grazia Tarsisano, Laura Di Renzo, Massimo Federici, Massimiliano Caprio, Antonino De Lorenzo

**Affiliations:** 1Department of Human Sciences and Promotion of the Quality of Life, San Raffaele Open University, 00166 Rome, Italy; giovanni.aulisa@uniroma5.it (G.A.); daniele.mar16@yahoo.com (D.M.); massimiliano.caprio@uniroma5.it (M.C.); 2Department of Systems Medicine, University of Rome “Tor Vergata”, 00173 Rome, Italy; federicm@uniroma2.it; 3Independent Researcher, via Venezuela 66, 98121 Messina, Italy; gianlucarizzo@email.it; 4Department of Experimental Medicine, Sapienza University of Rome, Viale Regina Elena 324, 00161 Rome, Italy; mariagrazia.tarsitano@uniroma1.it; 5Section of Clinical Nutrition and Nutrigenomic, Department of Biomedicine and Prevention, University of Tor Vergata, via Montpellier 1, 00133 Rome, Italy; laura.di.renzo@uniroma2.it (L.D.R.); delorenzo@uniroma2.it (A.D.L.); 6Laboratory of Cardiovascular Endocrinology, IRCCS San Raffaele Pisana, 00166 Rome, Italy

**Keywords:** TMAO, trimethylamine N-oxide, foods, fish, meat, eggs, dairy, microbiota

## Abstract

Introduction: Trimethylamine N-oxide (TMAO) may play a key mediator role in the relationship between the diet, gut microbiota and cardiovascular diseases, particularly in people with kidney failure. The aim of this review is to evaluate which foods have a greater influence on blood or urinary trimethylamine N-oxide (TMAO) levels. Methods: 391 language articles were screened, and 27 were analysed and summarized for this review, using the keywords “TMAO” AND “egg” OR “meat” OR “fish” OR “dairy” OR “vegetables” OR “fruit” OR “food” in December 2020. Results: A strong correlation between TMAO and fish consumption, mainly saltwater fish and shellfish, but not freshwater fish, has been demonstrated. Associations of the consumption of eggs, dairy and meat with TMAO are less clear and may depend on other factors such as microbiota or cooking methods. Plant-based foods do not seem to influence TMAO but have been less investigated. Discussion: Consumption of saltwater fish, dark meat fish and shellfish seems to be associated with an increase in urine or plasma TMAO values. Further studies are needed to understand the relationship between increased risk of cardiovascular disease and plasma levels of TMAO due to fish consumption. Interventions coupled with long-term dietary patterns targeting the gut microbiota seem promising.

## 1. Introduction

Trimethylamine N-oxide (TMAO) has been identified as an osmolyte molecule of the fish world [1]. In many marine species, it acts as a protein stabiliser that counteracts the concentration of urea. The oxidised form decomposes into trimethylamine (TMA) which is responsible for the characteristic odour of putrefied fish [2]. The free TMAO from seafood is directly absorbed into the systemic circulation without metabolism by the gastrointestinal microbiome. In humans, the gut microbiota is responsible for the formation of TMA [3]. The metabolic pathway allows the conversion of choline, betaine and carnitine molecules into TMA, whilst a liver enzyme belonging to the flavin-containing monooxygenases (FMO) family is responsible for its conversion into TMAO [4].

In recent years, numerous studies have suggested that high blood TMAO levels might be associated with heart disease, atherosclerosis, diabetes and cancer [5,6]. Compelling evidence suggests that circulating TMAO may promote atherosclerosis by altering the clearance of cholesterol in the liver, promoting inflammation and oxidation of LDL cholesterol by up-regulation of the macrophage scavenger and foam cell formation [7]. The current literature shows that TMAO may be a major risk factor for cardiovascular disease (CVD), especially in individuals who have already had a cardiovascular event or have a kidney disease [8,9].

The factors that can influence the plasma concentration of TMAO are different and complex [10]. For instance, TMAO precursors are among the main bacterial products produced by the intestinal microbiota, and it is likely that the microbiota is essential for TMAO metabolism [6].

The correlation between fish intake and TMAO has been demonstrated since the discovery of this molecule. For other foods, there is an extensive conflicting literature. The aim of our review is to evaluate all the studies in the literature to establish which foods are associated with an increase in plasma or urinary TMAO.

### Search Strategy and Selection Criteria

The present research was conducted and reported based on the PRISMA guidelines. We searched PubMed, Web of Science, EMBASE and Cochrane Central Register of Controlled Trials (CENTRAL) electronic databases using the following keywords as title/abstract fields: (“Trimethylamine N-oxide” OR “TMAO”) AND (“egg” OR “meat” OR “fish” OR “dairy” OR “vegetables” OR “fruit” OR “food”). We researched papers from 1 January 1990 to 1 December 2020. Studies published in languages other than English were not considered.

Figure 1 depicts the flow of the step-by-step process of applying the inclusion and exclusion criteria to generate a final number of studies for analysis in the review. Studies evaluating food intake in correlation with urinary or plasma TMAO were considered. Reviews, letters, comments, animal studies, and abstracts for posters were excluded. Thus, studies that focused on diets, supplements or nutraceutical use were not considered. Studies evaluating only TMAO measurement techniques or biochemical aspects without focusing on food were excluded.

391 studies with these characteristics were obtained; 35 papers were excluded because they were duplicate studies. Of the remaining 356, an additional 235 were excluded with use of the criteria above after a review of the title and abstract, leaving 121 articles for full text review. Ninety-four papers were not considered because they focused on diets, supplements or nutraceutical use. 27 papers were finally selected. The full texts of the available studies were analysed and the results are reported in this review.

## 2. Results

27 studies were considered for this review [11,12,13,14,15,16,17,18,19,20,21,22,23,24,25,26,27,28,29,30,31,32,33,34,35,36,37]. Their major features such as study design, methodology used for TMAO determination, and the effects of foods on TMAO are given in Table 1.

### 2.1. Fish

15 studies evaluated the possible relationship between fish and TMAO. 13 studies demonstrated that levels of urinary or plasma TMAO and/or TMA are significantly associated with the intake of fish [11,12,20,21,24,27,28,30,31,32,33,34,36]. One of the earliest studies evaluated 46 different foods and demonstrated that only fish and other sea-products significantly increase urinary TMAO [12]. Cho et al. [21] supposed that the rapid rise in circulating TMAO in response to fish consumption might demonstrate that the absorption of intact dietary TMAO is independent of the microbiota. In one paper, the association of fish and shellfish intake with TMAO plasma concentrations was limited to men only [31]. Only two papers demonstrated no association. Rohrmann et al. [17] revealed that meat, egg or fish consumption is not associated with TMAO, choline or betaine concentrations. Thus, a prospective cohort study showed that dietary intake of fish does not significantly impact the TMAO value in immunodeficient subjects [35].

Table 2 shows the four studies that specifically evaluated the influence on TMA and TMAO of eating different types of fish. TMAO was higher in subjects that eat saltwater fish and shellfish but not freshwater fish [27]. These data were confirmed in another study that evidenced higher TMAO values in the group that consumed shellfish and dark meat fish (tuna steak, mackerel, salmon, sardines, bluefish and swordfish) [36]. Another paper showed that cod intake has stronger effects on plasma and urine TMAO concentrations than salmon intake [33]. One study evaluated cooking methods and demonstrated that TMAO values are strongly associated with deep-fried fish consumption [27].

### 2.2. Eggs

17 studies that assessed the correlation between egg consumption and increased TMAO were evaluated. Robust correlations between TMAO and eggs were demonstrated in six studies [13,14,19,25,30,34]. Pignanelli et al. [25] described that the intake of egg yolk contributes significantly to plasma levels of TMAO. The phosphatidylcholine in egg yolk, via action of the intestinal microbiome, may be the major contributor to production of TMAO. The increase in TMAO induced by egg consumption has also been related to cardiovascular risk. Tang et al. studied the effects on plasma TMAO after a test dose of two hard-boiled eggs and found that patients in the highest quartile for TMAO had a 2.5-fold increase in the three-year risk of myocardial infarction or stroke [13]. In contrast, studies mainly starting from 2014 have assessed little or no effect of egg consumption on TMAO levels [12,15,17,18,21,22,23,27,31,35,37]. Some of these studies have revealed that plasma choline and betaine increase dose-dependently with egg intake. Many of these studies have been funded by the egg industry [15,18,21,22,23,37].

### 2.3. Meat

12 studies [12,17,20,21,26,27,28,30,31,34,35,36] evaluated the possible correlation between meat consumption and an increase of TMAO. Seven studies showed that the levels of urinary or plasma TMAO and/or TMA are significantly associated with the intake of meat. In most of these studies, the differences between white meat, red meat and preserved meat were not evaluated. Yu et al. showed that TMAO is strongly associated with deep-fried meat but not with red meat or poultry [27]. In one study, the effect on TMAO of meat and fish was compared. Fish was shown to induce a two-fold increase in urinary TMAO compared to meat [36]. Five papers [14,19,33,36,37] demonstrated that ingestion of meats has no measurable effects on plasma or urinary TMAO.

### 2.4. Dairy

Eight studies that assessed the correlation between fermented and non-fermented dairy consumption and increased TMAO were evaluated. Two papers [17,29] showed a positive association between dairy food consumption and plasma TMAO concentrations. Lower circulating and urinary TMAO with fermented dairy consumption compared to non-fermented dairy consumption has been shown [29].

### 2.5. Plant-Based Foods

Three papers evaluated plant-based food consumption [14,37]. Ingestion of fruits, vegetables, cereals [12] and fibre [35] did not significantly impact TMAO concentrations. Different foods were evaluated by Yu et al. [27]; they showed that consumption of soy foods or legumes does not modify urinary or plasma TMAO values.

## 3. Discussion

It has been supposed that TMAO may be atherogenic, prothrombotic and inflammatory [38]. Compared to healthy subjects, patients suffering from CVD have higher blood TMAO levels [39]. Thus, TMAO predicted CVD and mortality in a prospective cohort study [40]. CVD risk was related to blood serum TMAO levels, even after accounting for covariates such as meat, fish, cholesterol and energy intake. TMAO may be a cardinal halfway marker that connects dietary foods and fat with gut microbiota metabolism [41]. Recent studies have identified plasma TMAO as a dose-dependent risk factor for CVD that may promote atherosclerosis through the increase of cholesterol storage in macrophages [13,42,43]. TMAO is also related to metabolic syndromes and cancers [44]. However, from the review of the literature, there is no agreement whether TMAO may be a proatherogenic compound or, conversely, a marker of CVD. As proposed by Papandreou et al. [45], TMAO levels in CVD could be the result of disease-related dysbiosis and this would account for the observed intra-individual variability in TMAO levels. Furthermore, the microbiome that is implicated in TMAO metabolism may play an autonomous role in disease progression, making TMAO a biomarker and not an inducer of CVD. The production of TMAO could be an expression of individual differences in the gut microbiome [21].

A number of studies have provided hypotheses as to which processes induce systemic TMAO levels. Wang et al. [26] have suggested enhanced dietary precursors, increased microbial TMA/TMAO production from carnitine, and reduced renal TMAO excretion. They also revealed that ending consumption of red meat cuts down plasma TMAO within four weeks. Koeth et al. hypothesised [43] that in omnivores dietary L-carnitine is converted via a first fast generation of the atherogenic intermediate γ-butyrobetaine, succeeded by transformation into TMA via microbiota.

It is important to understand which foods or which other nutritional strategies (e.g., cooking method or selective microbiome intervention) have a greater effect on plasma TMAO levels, as this would allow for better nutritional prescribing, particularly in patients with modestly impaired renal function [9] and type-2 diabetes mellitus (T2DM) [46] (in which TMAO has been demonstrated to have even more deleterious effects on CVD risk). Some authors have proposed that a simpler method of reducing TMAO might be to eat fewer foods containing TMA precursors and increase those that favour non-TMA-producing bacteria (e.g., vegetables/fruit) or suppression of FMO3 activity (e.g., vegetables containing indole) [47].

Our review found that the majority of studies demonstrated a strong correlation between TMAO content (per gram of protein) and fish consumption. For other foods, especially those of animal origin, results are conflicting. Fruits, vegetables, cereals and fibre have been less investigated but would not appear to have any effect on TMAO. These data could be further evidence confirming that, in subjects with renal insufficiency, substituting animal-based proteins with plant-based proteins has shown reductions in the severity of hypertension, hyperphosphatemia and metabolic acidosis [48]. The majority of studies do not report a correlation between dairy consumption and TMAO. Consumption of fermented dairy products such as yogurt and cheese appears to have less effect on TMAO, probably because of the beneficial effect on gut microflora [29]. With regard to meat (white, red or processed) and eggs, many studies suggest a possible correlation with increased TMAO, but further independently conducted randomised controlled trials (RCTs) are needed to establish the full correlation. Nevertheless, compared to meat, fish has been shown to cause a two-fold increase in urinary TMAO.

Fish consumption induces a rapid rise in circulating TMAO because the absorption of intact dietary TMAO occurs independently of the gut microbiota [3]. Similar to that of omega-3 polyunsaturated fatty acids, the TMAO content in fish is different for each species (Table 2) and is mainly present in deep-sea varieties such as cod, haddock and halibut. The TMAO content of other fish, including salmon, depends on several factors including whether bred or wild and the period of capture [33].

How is it possible that fish consumption, which according to most studies reduces the risk of CVD [49], induces such a clear increase in plasma and urinary TMAO? Several hypotheses have been proposed to explain this apparent contradiction. A recent review by Ufnal M, et al. [50] postulated that increased plasma TMAO could be a compensatory effect that prevents cells from hydrostatic and osmotic stresses. The increase in plasma TMAO in CVD may be similar to that observed for plasma natriuretic peptide B, which is considered a marker of CVD risk but also a compensatory response that results in beneficial effects for the overloaded heart. Thus, a recent study in animal models has shown that a TMAO supplementation could even be beneficial in counteracting hypertension-related heart failure [51]. Similarly, in another recent review it was suggested that TMAO elevation may be a compensatory mechanism in response to disease. TMAO may act as a molecular chaperone to antagonise the disease progression [45]. The TMAO-fish paradox could be explained by the presence of heart-healthy nutrients in fish that would offset the negative effects of TMAO [52]. In fact, it would be wrong to focus only on the presence of TMAO and leave out other important elements. It has been seen that the omega-3 polyunsaturated fatty acids EPA and DHA are the basis of cardioprotective effects. These have an important metabolic action, reducing triglyceride values and improving the lipid profile through various mechanisms that have positive effects on cardiovascular health [53,54]. Practical implications also suggest that the method of preparing may influence the TMAO plasma level. Association of TMAO has been found with deep-fried meat or fish, but not with stir-fried meat or fish or deep-fried wheat or rice [27]. Elevated TMAO has also been linked to substantially increased risk for type 2 diabetes and metabolic syndrome. Indeed, as it has been demonstrated in models of ischaemic injury that the elevation of TMAO may be the result of injury caused by disease that would induce the up-regulation of TMAO by the FMO gene regulation [45].

DiNicolantonio et al. suggested that the correlations between TMAO and T2DM risk may be stronger than those for CVD risk. They also supposed that the increased risk for vascular events in subjects with elevated TMAO may be mediated largely by hepatic insulin resistance [55]. These assumptions could open a new scenario on TMAO as a marker associated with the intake of fish and meats. Thus, as we have shown in a previous review, higher intake of animal protein might be related to an increased risk of T2DM and CVD [56].

The variability in plasma and urine TMAO between foods could also be related to the methodology used to assess TMAO. As shown in Table 1, most papers evaluated only plasma TMAO without considering TMA or urinary excretion. As proposed by Papandreou et al. it would probably be more useful to use plasma TMA/TMAO ratio, because it is a more accurate and standardized marker [45].

Finally, we acknowledge our narrative review suffers from some limitations, mainly resulting from the heterogeneity of studies analysed in terms of research design and characteristics of subjects included. A potential limitation of the present revision is the fact that the search was limited to a few databases and only the English language; nevertheless, a large number of studies were identified. In order to find a correlation between individual foods and TMAO, we decided not to consider studies that included more than one food or dietary protocol. The decision to focus research on individual foods clearly restricted the results and thus the possibility of giving clear answers.

## 4. Conclusions

This review shows that most studies reveal a correlation between the consumption of many fish types (saltwater fish, dark meat fish and shellfish) and urine or plasma TMAO values. A correlation of increased TMAO with other animal-origin foods such as meat and eggs has been highlighted by most of the studies considered.

The food choice, whilst certainly very important, cannot be considered as the only factor for the total control of blood TMAO values. Gut microbiomes act directly and indirectly on the metabolism of TMAO and its precursors. In this sense, interventions targeting the gut microbiota seem promising.

Further studies, independent of the food industry, are needed to establish more definitive correlations between foods, microbiota and TMAO levels. 

## Figures and Tables

**Figure 1 nutrients-13-01426-f001:**
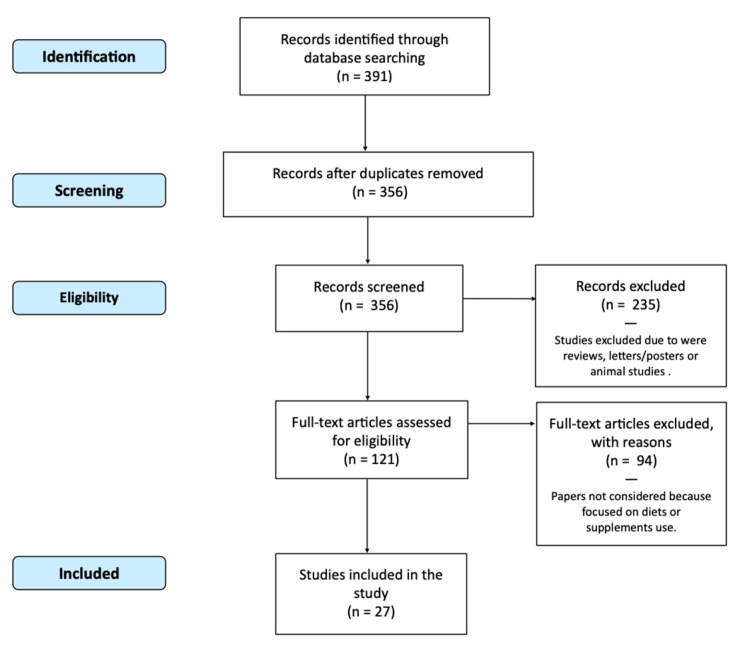
Flow Chart (PRISMA) of Studies Included.

**Table 1 nutrients-13-01426-t001:** Differences of urinary or plasmatic trimethylamine (TMAO) production from foods following human ingestion.

First Author	Year	Study Design	Sample	Methodology for TMAO Determination $	Red Meat (e.g., Beef, Lamb)	Meat Products (e.g., Sausages, Bacon)	White Meat (e.g., Chicken)	Fish	Eggs	Milk and Other Dairy Food	Plant-Based Foods	Reference	Funding
Svensson BG	1994	Comparative study	Urine	LC/MS				↑↑				[11]	Swedish Work Environment Fund and others
Zhang AQ	1999	Clinical trial	Urine	TMA/DMA	=	=	=	↑↑	=	=	=	[12]	The Leverhulme Trust
Tang WHW	2013	Prospective	Plasma and Urine	UHPLC-MS/MS					↑↑			[13]	National Institutes of Health and its Office of Dietary Supplements
Miller CA	2014	RCT	Plasma	LC/MS					↑↑			[14]	Egg Nutrition Centre
West AA	2014	Clinical trial	Plasma	UHPLC-ESI-MS/SM					=			[15]	American Egg Board and the Agriculture Research Institute at California State Polytechnic University, Pomona
Zheng H	2015	Cross-sectional	Urine	NMR						=		[16]	The Danish Council for Strategic Research, Arla Foods, and the Danish Dairy Research Foundation in the project
Rohrmann S	2016	Cross-sectional	Plasma	LC/MS	=		=	=	=	↑		[17]	Advancement of Human Nutrition
Di Marco DM	2017	Crossover randomised	Plasma	LC/MS					=			[18]	Egg Nutrition Centre
Iannotti LL	2017	RCT	Plasma	LC/MS					↑↑			[19]	The Mathile Institute for the Advancement of Human Nutrition
Kruger R	2017	Comparative study	Plasma	LC/MS	↑		↑	↑↑				[20]	Federal Ministry of Food and Agriculture
Cho CE	2017	RCT	Urine and Plasma	LC-MS/MS	↑		↑	↑↑	=			[21]	Egg Nutrition Centre and Beef Checkoff
Lemos BS	2018	Crossover randomised	Plasma	LC-MS/MS					=			[22]	Egg Nutrition Centre
Missimer A	2018	RCT	Plasma	LC/MS					=			[23]	Egg Nutrition Centre
Schmedes M	2018	RCT	Plasma	LC/MS				↑↑ §				[24]	Aarhus University project “Seafood protein in the prevention of the metabolic syndrome”
Pignanelli M	2019	Prospective cohort	Plasma	UHPLC-MS/MS					↑↑			[25]	Canadian Institutes of Health Research
Wang Z	2019	RCT	Plasma Urine	LC /MS	↑↑							[26]	National Institutes of Health and the Office of Dietary Supplements
Yu D	2019	Case-control multicentre	Urine	LC/MS	↑↑ #		↑↑ #	↑↑	=	=	=	[27]	National Institutes of Health (and others)
Andraos S	2020	Cross-sectional	Plasma	UHPLC-MS/MS	↑↑	= (children) ↑↑ (adults)	= (children) ↑↑ (adults)	↑↑		=		[28]	The New Zealand-Australia Life Course Collaboration on Genes, Environment, Nutrition and Obesity
Burton KJ	2020	Crossover randomised	Plasma and Urine	Plasma: UHPLC-MS/MS Urine: NMR (urine)						↑↑ (fermented) ç ↑ (non-fermented)		[29]	Joint Programming Initiative: A Healthy Diet for a Healthy Life
De Souza RJ	2020	Cross-sectional	Plasma	MSI-CE-MS	↑↑		↑↑	↑↑	↑↑			[30]	Canadian Institutes of Health Research (CIHR)
Gessner A	2020	Community-based	Plasma	LC/MS	=		=	↑↑ ^	=	=		[31]	None declared
Gibson R	2020	Cross-sectional	Urine	NMR				↑↑				[32]	None declared
Hagen IV	2020	RCT	Plasma and Urine	UHPLC-MS/MS				↑↑				[33]	Bergen Medical Research Foundation
Hamaya R	2020	Retrospective	Plasma	UPLC-ESI-MS/MS	=			↑↑	↑			[34]	US Highbush Blueberry Council
Macpherson ME	2020	Prospective cohort	Plasma	UHPLC-MS/MS	=		=	=	=	=	=	[35]	South-Eastern Norway-Regional Health Authority
Yin X	2020	RCT	Urine	NMR	↑			↑↑				[36]	NutriTech and the European Research Council
Zhu C	2020	Crossover randomised	Plasma	LC/MS					=			[37]	Egg Nutrition Council

* Effects on urinary and/or plasmatic TMAO based on study conclusions. $ Please refer to Appendix A for an explanation of the acronyms. (↑↑) food is strongly associated with TMAO concentrations;(↑) food is positively associated with TMAO concentrations; (=) food does not significantly impact TMAO concentrations; CVD; cardiovascular disease; RCT; randomised controlled trial; ç yogurt and cheese; # only fried; § (lean); ^ only men.

**Table 2 nutrients-13-01426-t002:** Associations of different types of fish intake with TMAO.

First Author	Year	Study Design	Cod	Farmed Salmon	Halibut	Herring	Mackerel	Sardine	swordfish	Shellfish	Clam	Tuna	Trout	Ref
Zhang AQ #	1999	Clinical trial	↑↑ 5135.3		↑↑ 8230.2	↑↑ 4345		↑ 1424.1	↑ 2769.4	↑ 1562	= 377.1	= 301.8	= 495.2	[12]
Yu D	2019	Case-control multicentre	↑↑ *	↑↑ *	↑↑ *	↑↑ *	↑↑ *	↑↑ *	↑↑ *	↑↑ *	↑↑ *	↑↑ *	=	[27]
Hagen IV	2020	RCT	↑↑	↑										[33]
Hamaya R	2020	Retrospective	↑↑	↑?			↑↑	↑↑	↑↑	↑↑		↑↑		[34]

TMA and TMAO: ↑↑ > 4000; ↑ 1000–3999; = less than 1000 mmol/8 h; # urinary TMAO production from foods following human ingestion (227 g); * only fried; RCT: randomised clinical trial.

## Data Availability

The data used in this manuscript are publicly available from previous publications and fully disclosed in Table 1 and Table 2 of the manuscript.

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
