# Peer review of "Association of Urinary and Plasma Levels of Trimethylamine N-Oxide (TMAO) with Foods"

_nutrients, 2021, doi:10.3390/nu13051426_

Round 1

Reviewer 1 Report

The manuscript by Lombardo et al. presents a review investigating the association between intake of specific food items (of animal origin) and TMAO levels. The study is based on a literature survey, and the approach for literature search is described and appears to be thoroughly and systematic. The literature survey reveals that many studies have reported strong associations between fish intake and TMAO levels, even though the TMAO level may vary according to fish species/type of shellfish. The reported associations between intake of other products of animal origin (meat, dairy and egg) and TMAO are much less convincing. As such, this is no surprise as many researchers have already discovered these facts. In fact, it has resulted in a controversy that many speculate about: how come that TMAO is much stronger linked with fish consumption, which is considered to provide us with healthy unsaturated fat, than with red meat intake, where TMAO has been postulated to represent a causal relationship to CVD? It seems the authors of this review ignore this controversy. It would be relevant with a more objective discussion where it is also questioned whether TMAO actually does have a causal effect on CVD? Or is it more a marker linked with red meat intake than a causal agent? Would it not be relevant to identify this instead of performing ‘more of the same’ RCTs? I think the authors should be more critical in their discussion of TMAO. There is a good review by Papandreou et al. published in Nutrients (2020, 12, 1330; doi:10.3390/nu1205133) that the authors are suggested to read and find inspiration from.

Author Response

How come that TMAO is much stronger linked with fish consumption, which is considered to provide us with healthy unsaturated fat, than with red meat intake, where TMAO has been postulated to represent a causal relationship to CVD? It seems the authors of this review ignore this controversy. It would be relevant with a more objective discussion where it is also questioned whether TMAO actually does have a causal effect on CVD? Or is it more a marker linked with red meat intake than a causal agent? Would it not be relevant to identify this instead of performing ‘more of the same’ RCTs?

We thank the reviewer for the valuable comment. We have reorganised the discussion by adding a section on the "fish-TMAO-CVD paradox" and a paragraph on the possible role of the microbiome.  We have also added, both in Table 1 and in the supplementary material, information on the methodology used by the different studies to assess TMAO. This could also influence the diversity in TMAO values between the different papers. 

I think the authors should be more critical in their discussion of TMAO. There is a good review by Papandreou et al. published in Nutrients (2020, 12, 1330; doi:10.3390/nu1205133) that the authors are suggested to read and find inspiration from.

We thank the reviewer for the interesting review. We have read the review and used it for editing the discussion of our paper. 

Reviewer 2 Report

In this review article the authors describe which foods have greater influence on the levels of TMAO in plasma and urine. It is very well known how fish and red meat contribute to have greater effects on the its levels. This is well written review, however it does not add a lot of new information to the existing literature. 

Introduction: The information provided for specific liver enzymes in the metabolism for TMAO has no relevance in regards to the content of the review article. 

The major limitation in this article is that there is no discussion about the  methodology used for the determination of TMAO in different articles quoted. 

Discussion: The authors should give more information if there is a direct relationship in increase in TMAO levels due to fish and cardiovascular diseases. This would help in understanding if consumption of fish is beneficial.

Overall this article does not add any novel information, but it looks like a good compilation of the all the references listed. 

Author Response

Introduction: The information provided for specific liver enzymes in the metabolism for TMAO has no relevance in regards to the content of the review article. 

We thank the reviewer for the helpful comment. We have removed the part about liver enzymes that has little relevance for the review article.

The major limitation in this article is that there is no discussion about the  methodology used for the determination of TMAO in different articles quoted. 

We thank the reviewer for the valuable suggestion that allowed us to improve the paper. We have added in Table 1 the Sample used (urine or plasma or both) and the methodology for TMAO determination. In the supplementary material (Table 1S) you can find a description of the different acronyms used. 

Discussion: The authors should give more information if there is a direct relationship in increase in TMAO levels due to fish and cardiovascular diseases. This would help in understanding if consumption of fish is beneficial.

We thank the reviewer for the valuable comment. We have reorganised the discussion by adding a section on the "fish-TMAO-CVD paradox" and a paragraph on the possible role of the microbiome.

Round 2

Reviewer 1 Report

The authors have considered and addressed the issues raised by the reviewers, and I think the manuscript can be accepted for publication now.